# The Heat Stress Transcription Factor LlHsfA4 Enhanced Basic Thermotolerance through Regulating ROS Metabolism in Lilies (*Lilium*
*Longiflorum*)

**DOI:** 10.3390/ijms23010572

**Published:** 2022-01-05

**Authors:** Chengpeng Wang, Yunzhuan Zhou, Xi Yang, Bing Zhang, Fuxiang Xu, Yue Wang, Cunxu Song, Mingfang Yi, Nan Ma, Xiaofeng Zhou, Junna He

**Affiliations:** 1Beijing Key Laboratory of Development and Quality Control of Ornamental Crops, College of Horticulture, China Agricultural University, Beijing 100193, China; cpwang@cau.edu.cn (C.W.); yunzhuanzhou@cau.edu.cn (Y.Z.); yangxi1994@hotmail.com (X.Y.); zhangbing0405@163.com (B.Z.); xufuxiang@cau.edu.cn (F.X.); wangyue10@cau.edu.cn (Y.W.); S20193172475@cau.edu.cn (C.S.); ymfang@cau.edu.cn (M.Y.); ma_nan@cau.edu.cn (N.M.); 2Key Laboratory of East China Urban Agriculture, Ministry of Agriculture and Rural Affairs, Institute of Leisure Agriculture, Shandong Academy of Agricultural Sciences, Jinan 250100, China

**Keywords:** lilium, heat stress, LlHsfA4, reactive oxygen species

## Abstract

Heat stress severely affects the annual agricultural production. Heat stress transcription factors (HSFs) represent a critical regulatory juncture in the heat stress response (HSR) of plants. The HsfA1-dependent pathway has been explored well, but the regulatory mechanism of the HsfA1-independent pathway is still under-investigated. In the present research, HsfA4, an important gene of the HsfA1-independent pathway, was isolated from lilies (*Lilium longiflorum*) using the RACE method, which encodes 435 amino acids. LlHsfA4 contains a typical domain of HSFs and belongs to the HSF A4 family, according to homology comparisons and phylogenetic analysis. *LlHsfA4* was mainly expressed in leaves and was induced by heat stress and H_2_O_2_ using qRT-PCR and GUS staining in transgenic *Arabidopsis*. LlHsfA4 had transactivation activity and was located in the nucleus and cytoplasm through a yeast one hybrid system and through transient expression in lily protoplasts. Over expressing *LlHsfA4* in *Arabidopsis* enhanced its basic thermotolerance, but acquired thermotolerance was not achieved. Further research found that heat stress could increase H_2_O_2_ content in lily leaves and reduced H_2_O_2_ accumulation in transgenic plants, which was consistent with the up-regulation of HSR downstream genes such as *Heat stress proteins* (*HSPs*), Galactinol synthase1 (*GolS1*), *WRKY DNA binding protein 30* (*WRKY30*), *Zinc finger of *Arabidopsis* thaliana 6* (*ZAT6*) and the ROS-scavenging enzyme *Ascorbate peroxidase 2* (*APX2*). In conclusion, these results indicate that LlHsfA4 plays important roles in heat stress response through regulating the ROS metabolism in lilies.

## 1. Introduction

Heat stress is an abiotic stress that plants often encounter due to global warming; the appropriate temperature for normal plant growth and development is surpassed, causing irreversible damage to plants and reducing crop yield. In the long-term evolutionary process of resisting heat stress, plants have evolved a complex network of signal perception, transmission and transcriptional regulation in the molecular response to heat stress. Upon receiving heat stress signals, plant plasma membrane enhances its fluidity, promoting Ca^2+^ influx and forming primary signals that are transmitted downward through protein phosphorylation to heat stress transcription factors (HSFs), which activate the expression of downstream genes such as heat stress proteins and antioxidants to alleviate plants exhibiting tolerance to heat stress [1,2,3]. Reactive oxidative species (ROS) that act as a second messenger are also induced in heat stress response [4,5].

HSF genes are present in all eukaryotes, with only one or a few in yeast and animals, but dozens are present in plants, indicating the structural complexity and functional redundancy of plant HSFs [6]. When subjected to heat stress, plant HSFs specifically bind to the heat stress element (HSE) of heat stress downstream gene promoters and recruit other proteins to form transcriptional complexes to regulate their expression and response to heat stress [7]. The structure of plant HSFs are conserved and include several important domains such as the DNA binding domain (DBD), the oligomerization domain (OD), the nuclear localization signal (NLS), the nuclear export signal (NES) and the transcriptional activation domain (AHA) and OD domain contains hydrophobic repeat (HR) regions A/B which are required for oligomerization [8]. HSFs were classified into three classes, HSFA, HSFB and HSFC, according to the peculiarities of the OD, which mainly refers to the numbers of amino acid residues between the HR-A and HR-B parts [6]. HSFAs have AHA domains and transcriptional activation activity and HSFBs lack AHA domains and do not have transcriptional activation activity, which may interact with HSFAs as repressors in *Arabidopsis* and tomatoes [9,10,11,12]. HSFCs have been reported to play roles in different abiotic stresses such as salt stress and development in rice and wheat [6,13,14,15]. HSFAs play a major role in heat stress response and the number of HSFAs is the largest among these three classes.

Studies have shown that HSFA genes are central regulators of heat stress response in plants [6]. In model and crop plants such as *Arabidopsis*, tomato and soybean, HsfA1 expresses constitutively in a normal condition and elevates quickly due to heat stress and it plays critical roles mainly in basal heat tolerance [16,17,18,19,20]; *HsfA2* and *HsfA3* are induced by heat stress and regulated by HsfA1, which could be key regulators in the establishment of acquired thermo-tolerance in plants [21,22,23]. HsfA1s also activate *HsfA7*, *HsfBs*, *Multiprotein bridging factor 1c* (*MBF1c*) and *dehydration responsive element-binding protein 2a* (*DREB2a*), which modulate the synthesis of heat stress response genes [19,24]. These genes are all regulated by HsfA1 and composed of HsfA1-dependent pathways in heat stress response, while other genes regulate thermotolerance independently of HSFA1, such as *HsfA4*, *HsfA5* and *HsfA9* [3,25,26]. Studies in different species suggest that HsfA4 may be involved in salt stress and heavy metal stress via ROS [27,28,29].

Plant ROS are chemically reactive oxygen metabolites and their derivatives, which are more reactive than oxygen atoms, mainly including hydrogen peroxide (H_2_O_2_), superoxide anions (O_2_^−^), hydroxyl radicals (HO^−^) and singlet oxygen (^1^O_2_). Heat stress induces large amounts of ROS in chloroplasts and mitochondria, which act as second messengers to activate heat stress response [4,5]. However, excessive ROS can damage plants, which cause a negative effect in plant responses to heat stress. *atrbohB* and *atrbohD* mutants, which are deficient in ROS-producing NADPH oxidizes in plant cell membranes, were sensitive to heat stress [1]. Ascorbate peroxidase (APX) and catalase (CAT) are two important classes of ROS scavenging enzymes and the double mutant *apx2 cat2* has also reduced thermo-tolerance [30]. Zinc finger proteins ZAT12, ZAT7 and ZAT10 regulated the expression of *APXs* in heat stress response [31,32]. *SNAC3* transcription factor in rice was induced by heat stress and reduced ROS damage by activating the expression of ROS scavenging enzyme genes [33]. These reports indicate that ROS is involved in plant response to heat stress through regulating ROS scavenging systems.

Lilies are colorful, fragrant flowers with a long growth period. The lily is one of four important cut flowers in the international flower market and is becoming increasingly popular in China [34,35,36,37]. Common cultivated cut lily species prefer a cool environment; for example, the suitable growth temperature of Oriental lilies is 25–28 °C during the day. Most regions in China have a continuous high temperature (more than 30 °C) in the summer that seriously affects lily growth and development, causing vanished buds and blind flowers, greatly affecting the quantity and quality of fresh cut lily flowers. Therefore, studying the molecular mechanism of heat stress response in lilies and breeding new heat-resistant lily varieties through genetic improvement is an effective way to solve this problem. We have focused on HSFAs and, from *Lilium longiflorum* ‘White Heaven’, have cloned *LlHsfA1*, *LlHsfA2a*, *LlHsfA2b*, *LlHsfA3A* and *LlHsfA3B*, which are all involved in heat stress response by the HsfA1-dependent pathway [20,22,23,38]. In this study, we characterized another HSFA gene *LlHsfA4*, which participates in the HsfA1-independent pathway in lilies. The expression of *LlHsfA4* was induced by heat and H_2_O_2_. Over expressing *LlHsfA4* in *Arabidopsis* improved the basic thermotolerance, reduced ROS content and up-regulated the expression of *AtAPX2*, which indicates that LlHsfA4 could play important roles in heat stress response through regulating the expression of ROS scavenging enzymes to control ROS levels in lilies.

## 2. Results

### 2.1. Molecular Cloning and Sequence Analysis of LlHsfA4

The *LlHsfA4* gene (Accession No.MZ970559) was cloned from *Lilium longiflorum* ‘White Heaven’ tissue culture seedlings by homologous cloning and rapid amplification of the cDNA ends (RACE) technique. The full-length sequence of LlHsfA4 contained a 5′-untranslated region (UTR) of 182 bp, a 3′-UTR region of 235 bp and an open reading frame (ORF) region of 1311 bp, encoding a 436 amino acid protein. Homology comparison of the amino acid sequence revealed that LlHsfA4 contains all the core domains of heat stress transcription factor, including the DNA binding domain (DBD), the oligomerization domain (OD), the nuclear localization signal (NLS), activator peptide motifs (AHA) and the nuclear export signal (NES) regions (Figure 1A). LlHsfA4 was highly similar to HsfA4 in all of the five domains in monocotyledonous plants such as *Elaeisguineensis*, *Phoenix dactylifera* and *Oryza sativa*, while it was less conserved in the C-terminal region compared with the same protein in dicotyledonous plants such as *Arabidopsis* and *Solanumly copersicum*.

LlHsfA4 was the closest genetic association with the class A4 according to phylogenetic analysis with 21 heat-stress transcription factors reported in *Arabidopsis*, which provided a further hint that LlHsfA4 was a novel HSF protein (Figure 1B).The phylogenetic tree was also analyzed using HsfA4s from other reported species, which showed that LlHsfA4 was the closest to HsfA4s in monocotyledonous plants such as *Asparagus officinalis*, *Elaeis guineensis*, *Phoenix dactylifera* and *Ananascomosus* (Appendix A). These analyses indicate that the cDNA cloned from lilies was the *HsfA4* gene and was named *LlHsfA4* to examine its function in heat stress response in Lilium.

### 2.2. Expression Analysis of LlHsfA4

To investigate the role of LlHsfA4 in heat stress response in lily plantlets under tissue culture, different expression patterns of *LlHsfA4* were analyzed, including organ expression and expression under heat stress. Organ expression was analyzed firstly and the organ mainly includes roots, bulbs and leaves. At normal temperature (25 °C), *LlHsfA4* expression was detected in all three organs and was highest in the leaves, compared to that in roots and bulbs (Figure 2A), suggesting that *LlHsfA4* constitutively expresses in all of the plantlets and mainly expresses in leaves that sense heat stress signal firstly. Thus, leaves were selected as the material to perform subsequent expression experiments.

In order to determine the expression pattern of *LlHsfA4* under heat stress, small plantlets of lilies were subjected to heat treatment at different temperatures for three hours. Compared with 25 °C, expression of *LlHsfA4* was enhanced at 28 °C, 32 °C, 37 °C and 42 °C, reaching a peak at 37 °C, which increased 4.7-fold (Figure 2B). Examination assay of mRNA levels of heat treatment at 37 °C with different times were then carried out. The expression of *LlHsfA4* increased first and then decreased with the extension of heat treatment time and the expression level was the highest at 3 h (Figure 2C).

As ROS are important second messengers in heat stress response and since heat stress induces ROS accumulation, the expression of *LlHsfA4* was detected in H_2_O_2_ treatment. Under H_2_O_2_ treatment, *LlHsfA4* expression was induced at 1 h and increased over 50-fold at 6 h, which indicates that *LlHsfA4* could be involved in the ROS pathway in heat stress response (Figure 2D).

To further identify the expression pattern and regulation of *LlHsfA4*, the promoter of *LlHsfA4* was cloned using a high-tail technique in lilies. About 1.8 kb of the genomic DNA sequence upstream of the *LlHsfA4* gene was isolated and analyzed for possible *cis*-elements using the PLACE data [39,40]. Unpredictably, there were no conserved HSE elements (AGAAnnTTCT) in the promoter, while *cis*-element associated hormone signals such as ABA-responsive element (ABRE), v-myb avian myeloblastosis viral oncogene homolog (MYB), anaerobic responsive element (ARE) and CGTCA-motif were found (Figure 3A). A pLlHsfA4-GUS binary vector was constructed and transformed into *Arabidopsis* and transgenic seedlings were then selected to determine activity of the *LlHsfA4* promoter using GUS staining. GUS activity was detected in all organs including leaves and roots; it was higher in leaves and hypocotyls and lower in roots, which indicates the constitutive expression activity of the *LlHsfA4* promoter (Figure 3B).GUS activity was also detected by different abiotic stresses such as heat, salt and osmosis (Figure 3C).The staining was higher only in heat stress, with no change under salt and osmotic stress compared with control, which means that *LlHsfA4* could respond to heat stress specially. GUS staining was deepened gradually under heat and H_2_O_2_ treatment at different times (Figure 3D,E). These results are consistent with the qRT-PCR experiment with lilies (Figure 2), which indicated that *LlHsfA4* expresses universally and is induced by heat and ROS.

### 2.3. LlHsfA4 Localized to the Nucleus and Exhibited Transactivation Activity

In order to further study the function of the HsfA4 protein, the subcellular localization of LlHsfA4 was investigated using two transient transformation assays. A binary vector containing LlHsfA4::eGFP driven by a super promoter was constructed. The fusion vector and a control vector were purified using a plasmid extraction kit and were transiently expressed in lily leaf protoplast cells using the PEG-Ca^2+^ method. As shown in Figure 4A, the LlHsfA4::eGFP signal was mainly localized in the nucleus, while the control GFP was localized in the nucleus and the plasma membrane. Another transient expression system of *N. benthamiana* was also used to confirm the localization. The vector was transformed into *Agrobacterium* and infected in *N. benthamiana* to observe the fluorescent signal. LlHsfA4::GFP was observed only in the nucleus and the green fluorescent was overlapped with the red fluorescent of nuclear localization marker NF-YA4-mcherry [41] (Figure 4B).While the control GFP was observed in the entire cell (Figure 4B). These results indicated that LlHsfA4 is a nuclear localization protein, which is consistent with its function of transcription factors.

Homologous proteins of HsfA4 in plants such as *Arabidopsis* [42], rice [43], wheat [44] and tomato [25] all have transcriptional activation activity. As LlHsfA4 is localized in the nucleus, it is speculated that LlHsfA4 may also have a transcriptional activation function. The activation activity of LlHsfA4 was detected by the yeast system. The results in Figure 5 show that all the transformed yeast could grow normally on a SD/-Trp plate. On the SD/-Trp-His select plate, the yeasts transformed with pBD-LlHsfA4 (FL) or pBD-GAL4 of the recombinant plasmids were able to grow normally, while the yeasts transformed with the negative control pGBKT7 plasmid did not. In the presence of X-Gal, yeast transformed with the pBD-LlHsfA4 (FL) and pBD-GAL4 recombinant plasmids turned blue within 8 h, while that transformed with pGBKT7 could not and the β-galactosidase activity of pBD-LlHsfA4 was much higher than that of pBD-GAL4 (Figure 5B). The results show that LlHsfA4 had transcriptional activation activity.

In order to further determine the transcriptional activation region of LlHsfA4, four short truncations of this protein were constructed in units of 50 amino acids from the C-terminal, including D1 (370 amino acid), D2 (320 amino acid), D3 (270 amino acid) and D4 (220 amino acid), none of which contain an AHA motif (Figure 5A). All of the short truncations were constructed into a pGBKT7 vector and performed a yeast one hybrid assay. The results in Figure 5B, D1 and D2 show transcriptional activation activity, while D3 and D4 had no transcriptional activation activity in X-Gal staining, which means that there is a transcriptional activation region was located between 320th and 370th amino acids. The activity of β-galactosidase was further measured to confirm this result. The transcriptional activity of D1 and D2 was significantly decreased compared to that of the full-length protein, which may be related to the deletion of the AHA motif in D1 (370th amino acid to the terminator) and this is consistent with predictions and previous reports. Another decrease occurred between D2 and D3. There may be another transcription activation domain located in the region between 270th and 320th amino acids, which were not predicted (Figure 5B). These results indicated two transcription activation domains in LlHsfA4: one is the predicted AHA in the C-terminal and the other is in the region between 270th and 320th amino acids.

### 2.4. Overexpression of LlHsfA4 Enhanced the Basal Thermotolerance of Transgenic *Arabidopsis* Plants

To test the function of LlHsfA4 in heat stress, *LlHsfA4* under the control of the super promoter was transformed into *Arabidopsis thaliana* wild-type Col by the *Agrobacterium*-mediated floral-dip method. Transgenic lines were screened by Hygromycin and the expression of *LlHsfA4* in transgenic lines was detected by qRT-PCR. Two independent lines, A4–12 and A4–43, were chosen for heat stress phenotypic observation with a high expression level (Figure 6A). Firstly, the phenotypes of the basal thermotolerance of the transgenic plants were examined using 5-day old seedlings that were exposed to 45 °C for 70 min and transferred to a normal growth environment (22 °C). After 7 days, two transgenic lines survived well, while almost of the wild-type Col died (Figure 6B). The survival rates of two transgenic lines were over 50%, which is significantly higher those of the wild type (Figure 6C). An acquired thermotolerance assay was then carried out. With different times of 45 °C treatment, there were no differences between the transgenic lines and the wild type (Appendix A), which means that LlHsfA4 was not involved in the acquired thermotolerance. These assays implied that LlHsfA4 plays important roles in the basal thermotolerance process.

In order to search the molecular mechanism of LlHsfA4, improving the basal thermotolerance of transgenic *Arabidopsis*, the expression of heat stress response downstream genes in transgenic lines was detected by qRT-PCR (Figure 7). Four small HSP genes (*AtHsp19.9*, *AtHsp22*, *AtHsp25.3* and *AtHsp17.6*), *AtGolS1*, *AtMBF1c*, *AtWRKY30*, *AtZAT6* and *AtZAT12* were examined in WT and transgenic lines. The expression of heat stress-related genes increased in transgenic lines compared with the wild type (Figure 7) and it was lower in Line A4-12, while the expression in Line A4-43 was higher, which is consistent with the expression of *LlHsfA4* in the transgenic lines (Figure 6A). This result indicated that the expression of HSR downstream genes was positively correlated with the expression level of *LlHsfA4* in transgenic lines, which enhanced the heat stress phenotype of transgenic *Arabidopsis*.

### 2.5. Accumulation of H_2_O_2_ Was Reduced in LlHsfA4 Transgenic *Arabidopsis* Plants

In the transgenic lines of *LlHsfA4*, the expression of *AtZAT6* was higher than that of the wild type, while the expression of *AtZAT12* did not change. AtZAT6 and AtZAT12 were required for the expression of *APXs* in heat stress response in the existing reports. Thus, it could be speculated that heat stress could promote the accumulation of H_2_O_2_ in lilies according to the above result. Firstly, H_2_O_2_ content was detected by DAB staining in ‘White Heaven’ tissue culture seedlings (Figure 8A) and the content of H_2_O_2_ was accumulated with 37 °C treatment, which is similar to the result of H_2_O_2_ treatment. This result was verified with the discs from mature leaves of cut flowers ‘Siberia’ (Figure 8B) and the heat stress and H_2_O_2_ treatment could enhance the H_2_O_2_ content in lily plants. In addition, superoxide anions (O_2_^−^) was examined by NBT staining in ‘White Heaven’ tissue culture seedlings (Figure 8A), which content was increased with 37 °C treatment and H_2_O_2_ treatment. These result indicated that ROS could be involved in heat stress response.

The binary vector containing *LlHsfA4* under the control of the super promoter was transformed into leaf discs for transient expression and 37 °C treatment induced H_2_O_2_ content in the control discs, while H_2_O_2_ content reduced in discs over expressing *LlHsfA4* (Figure 8C), which means that the accumulation of H_2_O_2_ was inhibited with the appearance of LlHsfA4. To confirm this result, leaves of transgenic plants were used for DAB staining. At heat (37 °C) and H_2_O_2_treatment, H_2_O_2_ content was decreased in all transgenic lines (Appendix A), which was consistent with the transient expression. There are many ROS-related scavenging enzymes, including eight ascorbate peroxidases (*AtAPXs*). They were used to examine their expression to identify which ones could be regulated by HsfA4; the result showed that only the expression of *AtAPX2* was higher in the two transgenic lines than that of the wild type (Figure 9 and Appendix A). These results indicated that HsfA4 could regulate the expression of *APX2* to control the ROS level and thus improve heat stress of plants.

We can conclude that LlHsfA4 may enhance the basal thermotolerance of *Arabidopsis* by directly increasing the expression of heat stress response genes and may improve the basal thermotolerance of *Arabidopsis* by reducing the accumulation of H_2_O_2_.

## 3. Discussion

### 3.1. LlHsfA4 Is a New HSF Transcription Factor in Lilies

Plants have evolved a variety of molecular mechanisms in response to heat stress which affects the growth and development of plants. HSFs are important regulatory factors in heat stress response and they are divided into HsfA1-dependent and HsfA1-independent groups [3]. In the previous study, we cloned *LlHsfA1*, *LlHsfA2a*, *LlHsfA2b*, *LlHsfA3a* and *LlHsfA3b* from lilies [20,22,23,38] and they were all involved in thermotolerance and belong to the HsfA1-dependent group. In this article, we cloned *LlHsfA4*, encoding a new HSF transcription factor belonging to the HsfA1-independent pathway. LlHsfA4 has conserved domains of HSFs such as DBD and OD in the N-terminus, NLS in the middle and AHA and NES in the C-terminus (Figure 1). The subcellular localization assay and transactivation assay showed that LlHsfA4 was mainly localized in the nucleus and had transcriptional activation activity with two transcription activation domains (Figure 4 and Figure 5). These results suggest that LlHsfA4 is a new HSF transcription factor in lilies.

The promoter of *LlHsfA4* was cloned and analyzed using GUS staining and the result showed that *LlHsfA4* expressed in all organs, was higher in leaves and hypocotyls and was lower in roots (Figure 3B or Figure 2A) and induced by heat in leaves more than roots, which is different from the *PuHsfA4* induced in *Populus* roots by zinc [28]. The expression of *LlHsfA4* was significantly induced by heat, but it did not increase in response to NaCl and mannitol stresses (Figure 3C), which means that promoter of *LlHsfA4* does not respond to salt stress as the homologous gene in *Arabidopsis* and Chrysanthemum [27,45]. While the over expressing *LlHsfA4* lines grew better than the wild type under 200 mM NaCl treatment (Appendix A), which indicates that LlHsfA4 could response to salt stress that need more assays to identify the mechanism. This difference could be explained by the different kinds of plant families. Additionally, no HSE elements were found in the promoter of *LlHsfA4* (Figure 3A), which was similar to the promoter of *AtHsfA4* in *Arabidopsis*, meaning that the expression of *HsfA4* could not be regulated by HsfA1, consistent with the classification of HsfA4 to HsfA1-independent transcription regulation networks [3,46].

The promoter of *LlHsfA4* was cloned and analyzed using GUS staining and the result showed that *LlHsfA4* expressed in all organs, was higher in leaves and hypocotyls and was lower in roots (Figure 3B or Figure 2A) and induced by heat in leaves more than roots, which is different from the *PuHsfA4* induced in *Populus* roots by zinc [28]. The expression of *LlHsfA4* was significantly induced by heat, but it did not increase in response to NaCl and mannitol stresses (Figure 3C), which means that promoter of *LlHsfA4* does not respond to salt stress as the homologous gene in *Arabidopsis* and Chrysanthemum [27,45]. While the over expressing *LlHsfA4* lines grew better than the wild type under 200 mM NaCl treatment (Appendix A), which indicates that LlHsfA4 could response to salt stress that need more assays to identify the mechanism. This difference could be explained by the different kinds of plant families. Additionally, no HSE elements were found in the promoter of *LlHsfA4* (Figure 3A), which was similar to the promoter of *AtHsfA4* in *Arabidopsis*, meaning that the expression of *HsfA4* could not be regulated by HsfA1, consistent with the classification of HsfA4 to HsfA1-independent transcription regulation networks [3,46].

### 3.2. LlHsfA4 Plays Important Roles in Basal Thermotolerance

HSFs are known to regulate heat stress response in different plants species and the expression of *LlHsfA4* was induced by heat. LlHsfA4 could be involved in the adapting process in high-temperature conditions in lilies. In this study, the function of LlHsfA4 was analyzed by over-expressing *LlHsfA4* in *Arabidopsis*. Transgenic *Arabidopsis* lines grew to a greater extent than the wild type and the survival ratio was higher than that of the wild type under basal thermo-tolerance treatment (Figure 6), while there were no differences between the transgenic lines and wild-type in the acquired thermotolerance assay (Appendix A). These heat treatment assays indicated that LlHsfA4 plays important roles in heat stress responses in lilies, especially in basal thermo tolerance. 

Downstream genes of HSFs such as *AtGolS1*, *AtMBF1c*, *AtWRKY30*, *AtZAT6* and *AtZAT12* were examined in this study. The expression of almost these genes increased in the over expressing *LlHsfA4* lines, which is consistent with the phenotype of the basal thermo-tolerance of the transgenic lines, indicating that LlHsfA4 could enhance heat tolerance by regulating the expression of these genes. In *Arabidopsis*, the expression levels of *HSP17.6*, *ZAT6*, *ZAT12* and *WRKY30* are much higher in *AtHsfA4* over-expressing lines than those in the wild type and heat treatment reduces this expression difference [47]. It has also been reported that a high temperature and salt can promote HsfA4 binding to the promoters of *HSP17.6*, *ZAT12* and *WRKY30*, respectively, and the combined action of high temperature and salt increased these bindings [45]. These results suggest that HsfA4 is involved in different abiotic stress responses by regulating the same downstream gene expression.

### 3.3. LlHsfA4 Could Regulate ROS Levels through Inducing the Expression of APX

HsfA4 plays important roles in salt stress, oxidative stress and heavy metal stress in several reported studies, which may be related to the metabolism balance of ROS. In monocotyledonous plants, *TaHsfA4a* in wheat (*Triticum aestivum*) and its homologous gene *OsHsfA4a* in rice (*Oryza sativa*) were obtained by screening Cd tolerance genes, which participate in Cd tolerance by regulating expression of the metallothionein gene *MT-1*. Further experiments showed that Cd tolerance of HsfA4 was related to Ala 31 and Leu 42 in the DBD domain [44]. In *Populus*, *PuHsfA4* reduced ROS in roots by regulating the expression of glutathione-S-transferase U17 (*PuGSTU17*) and phospholipase A2 (*PuPLA2*), thereby activating the antioxidant system and promoting zinc tolerance [28]. In chrysanthemum, *CmHsfA4* reduces ROS levels by inducing ROS scavengers such as SODs, APXs and CATs [27]. In this study, H_2_O_2_ was detected in leaves of tissue culture seedlings and leaf discs of cut flower using DAB staining. High temperatures promoted the accumulation of H_2_O_2_ in lilies, which is similar to the phenotype of exogenous H_2_O_2_ treatment, indicating that ROS were important second messengers in the process of heat stress response. High temperature treatment reduced the H_2_O_2_ content of *Arabidopsis* transgenic lines of *LlHsfA4*. Similarly, high temperature treatment also reduced the H_2_O_2_ content in lily leaf discs with a transient over expression of *LlHsfA4*. These results are the same as those reported in previous studies, indicating that *LlHsfA4* may be involved in the degradation of ROS during heat stress responses in Lilies.

Subsequently, we detected the gene expression of enzymes related to ROS metabolism in transgenic lines, such as *APXs*, *SODs*, *CATs* and *GRs*. Most genes did not change their expression in transgenic lines, but *APX2* did (Figure 9 and Appendix A). There are eight isoforms of APXs in *Arabidopsis*, three cytosolic types (APX1, APX2 andAPX6), two chloroplast types (stoma sAPX and thylakoid tAPX) and three microsomal types (APX3, APX4 andAPX5) [48]. Among these eight genes, the expression of *APX2* was increased in all transgenic lines, while the expression of *APX1*, *APX3*, *APX5* and *APX6* was only increased in A4–43, which has the highest level of *LlHsfA4* in the two transgenic lines. The expression of *APX2* in transgenic lines is much higher than that in the wild type and the expression of other *APXs* is about twice of that in the wild type, which means that the expression of *APX* is related to the expression of *LlHsfA4*. A higher expression of *LlHsfA4* induced a higher expression of *APXs*, indicating that *APX2* may be a potential target gene of LlHsfA4. We tried to amplify the promoter of *APXs* and found that there was a long intron in the 5′ sequence of the genome DNA [49,50] and it was difficult to use Hi-tail PCR to obtain the promoter. The regulation mechanism of APXs by LlHsfA4 needs further research.

### 3.4. Complexity of Relationship between LlHsfA4 and ROS in Heat Stress Response

HsfA4 has been shown to participate in ROS signals by regulating the expression of metabolically related ROS genes. According to existing articles, HsfA4 mediates ROS metabolism by inducing *ZAT12* and *APX1*, of which ZAT12 regulates the expression of *APX1* as a transcriptional inhibitor [31,32,47]. Moreover, HsfA4 can bind to the promoter of *ZAT12* to activate *ZAT12* expression [45]. Our result showed that the expression of *ZAT6* and *APX2* was higher in *LlHsfA4* over expression lines than in the wild type (Figure 7 and Figure 9), indicating that LlHsA4 could regulate ROS degradation through ZAT6 and is located upstream of the ROS signals. Whether LlHsfA4 can directly regulate APX or through *ZAT6* regulating *APX* expression in lilies remains to be studied.

As second messengers, ROS have also been reported to play roles in responding to heat stress. Heat stress causes chloroplasts, mitochondria, peroxisomes and cell membranes to release ROS [46,51]. ROS from different cellular compartments can activate ANP1 (MAPK kinase kinase)—MAPK3/MAPK6 cascades to phosphorylate HsfA4a and activate the transcriptional activity of HsfA4a [45,47,52,53], indicating that the protein activity of HsfA4 is regulated by ROS. In our experiment, the expression of *LlHsfA4* is induced by H_2_O_2_ (Figure 2 and Figure 3) and the promoter of *LlHsfA4* contains ARE cis-acting elements, which is an anaerobic-induction-related cis-acting element related to hypoxia stress and flood hypoxia [54,55]. These results suggest that ROS can regulate the expression and protein activity of LlHsfA4 as a signal molecule.

Thus, the complex relationship between LlHsfA4 and ROS in lilies has been elucidated. When the content of ROS was excessive, as second messengers, ROS can transmit heat stress signals downward to activate *LlHsfA4* to regulate *APX* expression, which could degrade ROS to a controllable range and thus decrease the toxicity of those ROS. This feedback regulation not only ensures that lilies have sufficient energy to adjust their metabolism, but also ensures that these plants have an appropriate heat stress response.

## 4. Materials and Methods

### 4.1. Plant Materials and Growth Conditions

Longiflorum hybrid ‘White heaven’ (*Lilium longiflorum*) was cultured in a Murashige and Skoog (MS) medium at 22 °C over a 16 h light/8 h dark photoperiod in a culture room. *Arabidopsis thaliana* (Col-0) were used for genetic transformation and were grown over a 16 h light/8 h dark light photoperiod at 22 °C.

### 4.2. Gene Cloning and Sequence Analysis

Total RNA was extracted from ‘White heaven’ leaves using the RNAprep Pure Plant Kit (Tiangen, Beijing, China) according to the manufacturer’s instructions. After DNase I (Takara, Bio, Beijing, China) treatment, 1 μg of RNA was subject to a reverse transcription reaction using the HiScript Q RT SuperMix Kit (Vazyme, Nanjing, China). The degenerate primers were designed (Appendix A) and the partial conserved sequence of *LlHsfA4* was amplified by homology-based cloning.The full-length sequence of *LlHsfA4* was then obtained by 5′- and 3′- one step Full Race Kit (Takara, Japan). The conserved domain prediction was carried out using the DNAMAN software and NCBI (https://www.ncbi.nlm.nih.gov/, accessed on 28 August 2021) website. Thephylogenetic tree was analyzed using the ClustalW 2.0 and MEGA 5.0 software.

### 4.3. Gene Expression Assay

The roots, bulbs and leaves of 4-week-old lily plantlets were taken as materials. The RNA of different tissues was extracted and real-time quantitative PCR was used to detect the expression of *LlHsfA4*. To analyze the response of *LlHsfA4* to different abiotic treatments, lily plantlets were exposed to different temperatures (25, 28, 32, 37 and 42 °C) for 3 h or to different durations of heat stress (0, 1, 3 and 6 h) at 37 °C, or they were treated with 1 mM H_2_O_2_ for different durations (0, 1, 3 and 6 h). After treatment, the RNA of these leaves was extracted for expression analysis. *18S rRNA* was used as a quantification control, which has been validated in previous studies [23]. qRT-PCR analysiswas performed with the 2^−ΔΔCT^ method and primers designed for qPCR analysis are listed in Appendix A.

To analyze the expression levels of heat-related and ROS-related genes in wild-type and transgenic *Arabidopsis* plants, the 10-day-old seedlings were collected and the RNA was extracted for real-time quantitative PCR analysis. Each experiment included three biological replicates. All relevant primers are listed in Appendix A.

### 4.4. Promoter Isolation and GUS Activity Assay

Genomic DNA extraction was carried out using the Genomic DNA Extraction Kit (Tiangen, Beijing, China). The 1800-bp promoter sequence upstream of the ATG of *LlHsfA4* was obtained by the hiTAIL-PCR method [56].

For promoter activity research, the DNA of 1800bp promoter was amplified by PCR and cloned between the *Pst*I and *Xma*I sites of the pCAMBIA1391 vector, which contains the *GUS* gene. The *pHsfA4*::GUS vector was obtained and transformed into the *Agrobacterium tumefaciens* GV3101 strain and the *Arabidopsis* plants were then infected using the floral-dip method [57]. The transgenic seedlings were screened on the MS medium containing 30 mg/L hygromycin to obtain homozygous transgenic plants for further analysis.

Histochemical staining for the GUS activity assay in transgenic plants was performed following the methods of He et al. (2012) [58]. 10-day-old seedlings were treated with water (Con), a salt solution (NaCl, 150 mM), or a mannitol solution (300 mM) for 3 h, or 1 mM H_2_O_2_ for different durations (0, 1, 3 and 6 h) and with heat stress (37 °C) for 3 h before being subjected to GUS analysis. The treated seedlings were immersed in a staining solution and incubated at 37 °C for 12 h and they were then decolorized with 75% ethanol for 24 h.

### 4.5. Subcellular Localization of LlHsfA4

The ORF of *LlHsfA4* was amplified by primers with *Xba*I and *Kpn*I sitesandcloned into a pCAMBIA1300 vector modified with a super promoter and eGFP cDNA to construct *pSuper*::LlHsfA4-GFP, which was used for subcellular localization, the transient expression of lily leaves and transformation in *Arabidopsis*.

The binary expression vector *pSuper*::LlHsfA4-GFP was transformed into lily protoplasts and tobacco leaves to examine the subcellular localization of LlHsfA4. GFP images were obtained by a Zeiss LSM510 META confocal microscope with an excitation at 488 nm and an emission at 525 nm. Protoplast transformation was performed according the *Arabidopsis* protoplasts transient transform [58]. For tobacco injection, the binary vector was transformed into Agrobacterium GV3101 strains and were injected into *Nicotiana benthamiana* leaves with nuclear localization marker NF-YA4-mcherry and the silencing suppressor P19 [41] and the treated tobacco seedlings were then put in a chamber with a 16 h light/8 h dark light photoperiod at 22 °C. Pictures were obtained after 48 h–72 h [59].

### 4.6. Transcriptional Activity Analyses

The pGBKT7 vector was used for transcriptional activity analysis. Full length (FL) and four short truncated sequences (D1, D2, D3 and D4) of LlHsfA4 were amplified individually using primers with *Nde*I and *Bam*HI sites and then cloned into pGBKT7to construct pBD-LlHsfA4 (FL), pBD-LlHsfA4 (D1), pBD-LlHsfA4 (D2), pBD-LlHsfA4 (D3) andpBD-LlHsfA4 (D4) vectors. The primers used in these experiments are listed in Appendix A. The vectors of pBD-LlHsfA4 (FL), pBD-LlHsfA4 (D1), pBD-LlHsfA4 (D2), pBD-LlHsfA4 (D3), pBD-LlHsfA4 (D4), pBD-GAL4 (positive control) and pGBKT7 (negative control) were introduced into AH109 yeast cells. Screening was performed on an SD/-Trp medium and an SD/-Trp-His medium and β-Gal activity was measured via an enzyme assay [23].

### 4.7. Thermotolerancetest of Transgenic *Arabidopsis*

Recombinant vectors of pSuper::LlHsfA4-GFP were transformed into the *Agrobacterium tumefaciens* GV3101 strain and infected *Arabidopsis* plants using the floral-dip method [57]. The transgenic seedlings were then selected on a MS medium containing 30 mg/L hygromycin to obtain homozygous transgenic plants for a thermotolerance test.

For the thermotolerance assay, *LlHsfA4* over expressing plants A4–12 and A4–43 as well as wild-type *Arabidopsis* were examined. The 5-day-old seedlings were transferred to a 45 °C environment for 70 min and then transferred to a 22 °C normal growth condition for recovery culture. The plant status was observed, photographed and recorded.

### 4.8. Transient Expression of Lily Leaves

The recombinant pSuper::LlHsfA4-GFP vector was transferred into the *Agrobacterium tumefaciens* strain GV3101, placed in a liquid LB medium overnight, collected by centrifugation, re-suspended in an infiltration buffer (10 mM MgCl_2_, 200 mM acetosyringine and 10 mM MES, pH 5.6) to a final OD_600_ of about 1.0 and finally, placed in the dark for 3 h. Discs with a 1 cm diameter from lily leaves were excised using a hole-puncher, infiltrated with an infection solution using a vacuum at −0.9 M Pa, washed with deionized water, placed on a semi-solid plate (0.4% agar) at room temperature for 48 h in the dark and then used for DAB staining.

### 4.9. ROS Detecting

In order to detect the effect of LlHsfA4 on ROS accumulation, DAB and NBT staining were used to detect H_2_O_2_ and O_2_^−^ accumulation separately. The specific method was as follows: The processed plant materials were placed in a 1 mg/L DAB solutionsoaked for 8 h, or in a 1 mg/L NBT solution for 15 min and then placed in 75% ethanol to decolorize. Photos were then taken for observation purposes [58].

### 4.10. Statistical Analysis

*T*-test analysis of variance was employed to identify treatment means that differed statistically. GraphPad Prism v7.00 was used for all statistical analyses.

## 5. Conclusions

LlHsfA4 can regulate ROS metabolism to enhance basal thermo-tolerance during heat stress response in lilies.

## Figures and Tables

**Figure 1 ijms-23-00572-f001:**
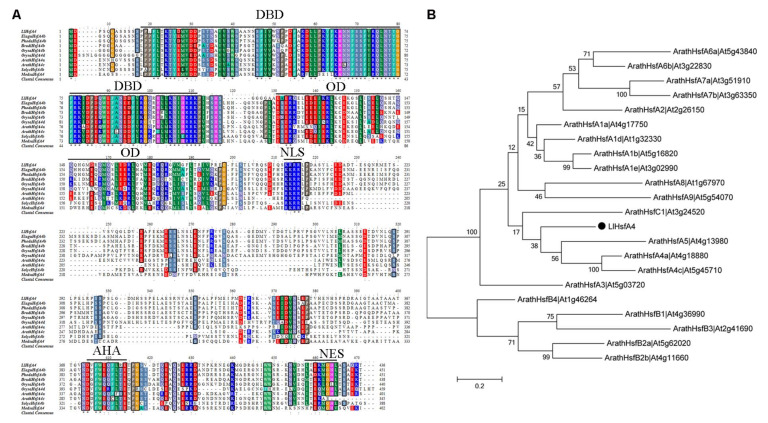
Alignment of amino acid sequence of HsfA4 in different plants. (**A**) Sequence alignment of LlHsfA4 with HsfA4 from *Elaeis guineensis*, *Phoenix dactylifera*, *Brachypodium distachyon*, *Oryza sativa*, *Arabidopsis*, *Solanumly copersicum* and *Medicago sativa*. The conserved DNA binding domain (DBD), the oligomerization domain (OD), the nuclear localization signal (NLS), activator peptide motifs (AHA) and the nuclear export signal (NES) are indicated by dark colored lines. (**B**) Phylogenetic tree of LlHsfA4 and all HSFs in *Arabidopsis*. This tree was constructed by ClustalW 2.0 and MEGA 5.0. The dark dot indicates an LlHsfA4 protein.

**Figure 2 ijms-23-00572-f002:**
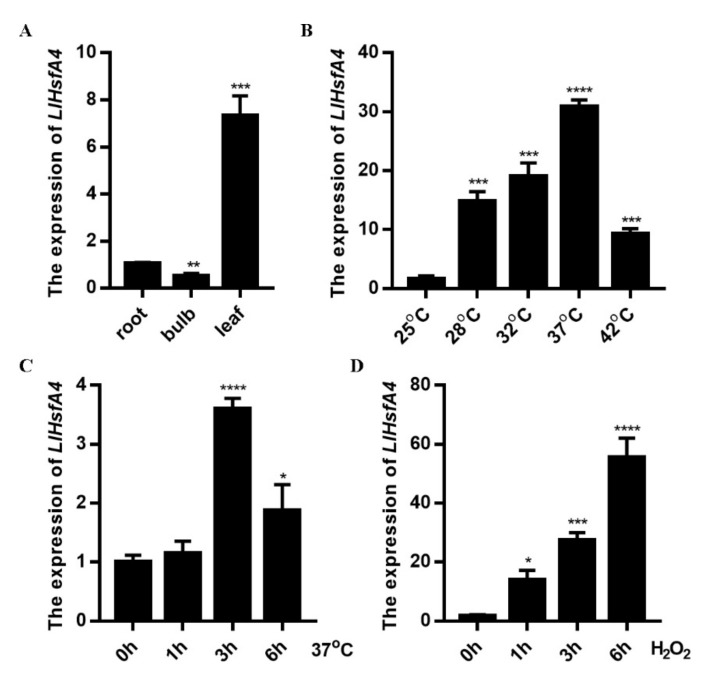
Expression analysis of HsfA4 in organs and heat stress condition by qRT-PCR. (**A**) Relative expression of LlHsfA4 in roots, bulbs and leaves. (**B**) Lily seedlings were treated with heat temperatures of 28 °C, 32 °C, 37 °C and 42 °C. (**C**) Lily seedlings were treated using 37 °C for 1 h, 3 h and 6 h. (**D**) Lily seedlings were treated using H_2_O_2_for 1 h, 3 h and 6 h. 18S rRNA was used as a control. Three independent experiments were performed each with three technical replicates, showing one experiment result. *t*-test analysis of variance was employed to identify treatment means that differed statistically. Samples with different letters are significantly different: * *p* < 0.05, ** *p* < 0.01, *** *p* < 0.001, **** *p* < 0.0001, Unmarked means non-significance.

**Figure 3 ijms-23-00572-f003:**
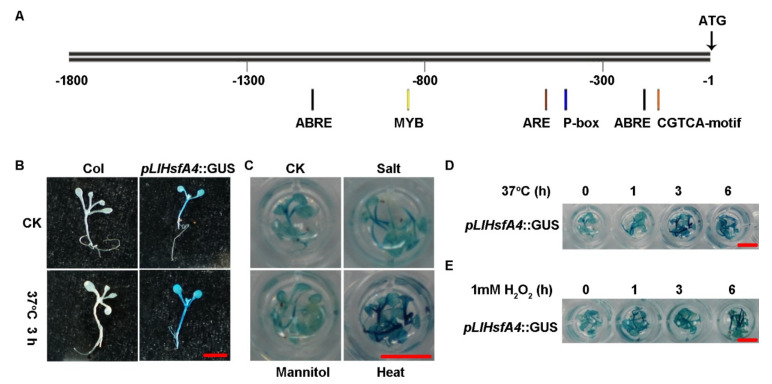
Expression analysis of *HsfA4* using *pLlHsfA4*::GUS transgenic *Arabidopsis*. (**A**) Cis-element analysis of *LlHsfA4* promoter using PLACE data. GUS staining of pLlHsfA4::GUS transgenic *Arabidopsis* seedlings were treated with different abiotic stresses: heat stress in (**B**–**D**), salt(150 mM NaCl) and osmosis (300 mM Mannitol) in (**C**) and ROS(1mM H_2_O_2_) in (**E**). Bar: 50 mm.

**Figure 4 ijms-23-00572-f004:**
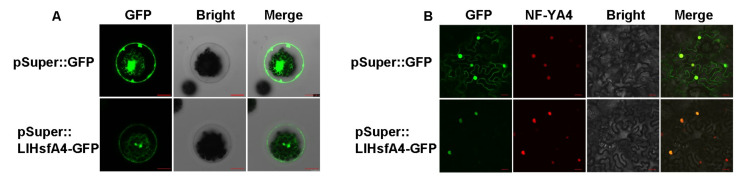
Subcellular localization of LlHsfA4. LlHsfA4-eGFP fusion protein was localized in the nucleus of a lily mesophyll protoplast (**A**) and *N. benthamiana* (**B**). Bar: 20 µm.

**Figure 5 ijms-23-00572-f005:**
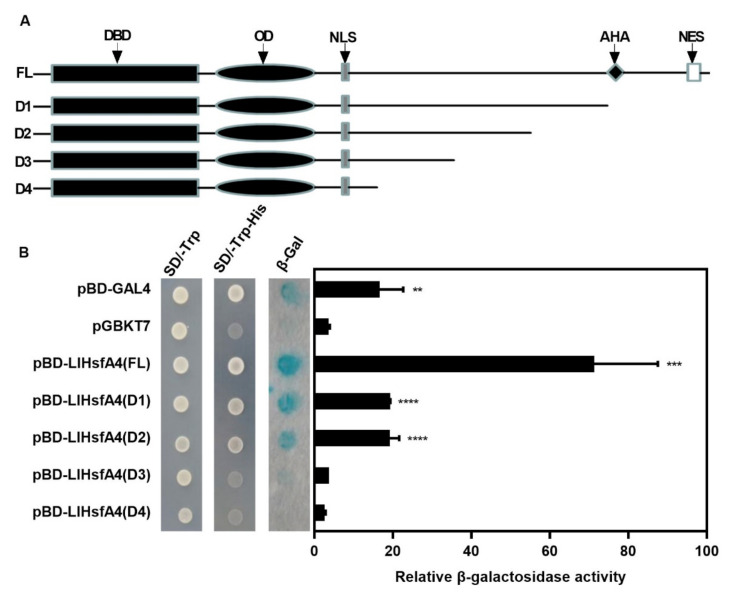
Transactivation activity of LlHsfA4 in the yeast strain AH109. (**A**) Different deletion forms of LlHsfA4 were constructed for the transcriptional activation assay. (**B**) Transactivation activity of different constructs were examined in yeast and the β-galactosidase activity of the transformed yeast cells was detected and measured. The transformed yeast cells were grown in a SD medium with or without histidine and X-gal staining was used for detecting β-gal activity. β-gal activity was measured through an enzyme assay. Three independent experiments were performed, each with three technical replicates, showing one experiment result. T-test analysis of variance was employed to identify treatment means that differed statistically. Samples with different letters are significantly different: ** *p* < 0.01, *** *p* < 0.001, **** *p* < 0.0001, Unmarked means non-significance.

**Figure 6 ijms-23-00572-f006:**
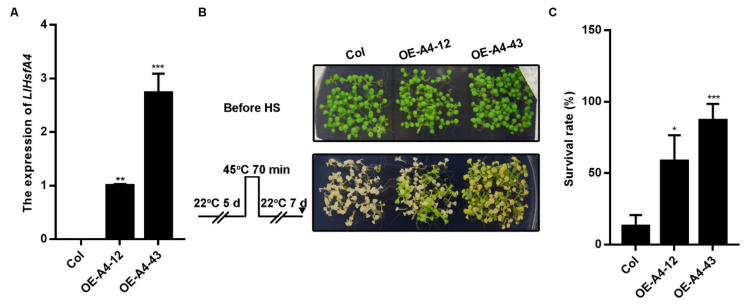
Transgenic *Arabidopsis* plants of *LlHsfA4* enhanced basal thermotolerance. (**A**) Overexpression of *LlHsfA4* in transgenic lines was examined using qRT-PCR. (**B**) Images of two transgenic lines and the wild type (Col) under normal conditions and recovery for 7 d after heat stress. (**C**) Survival rates of transgenic lines were calculated in (**B**). *t*-test analysis of variance was employed to identify treatment means that differed statistically.Samples with different letters are significantly different: * *p* < 0.05, ** *p* < 0.01, *** *p* < 0.001.

**Figure 7 ijms-23-00572-f007:**
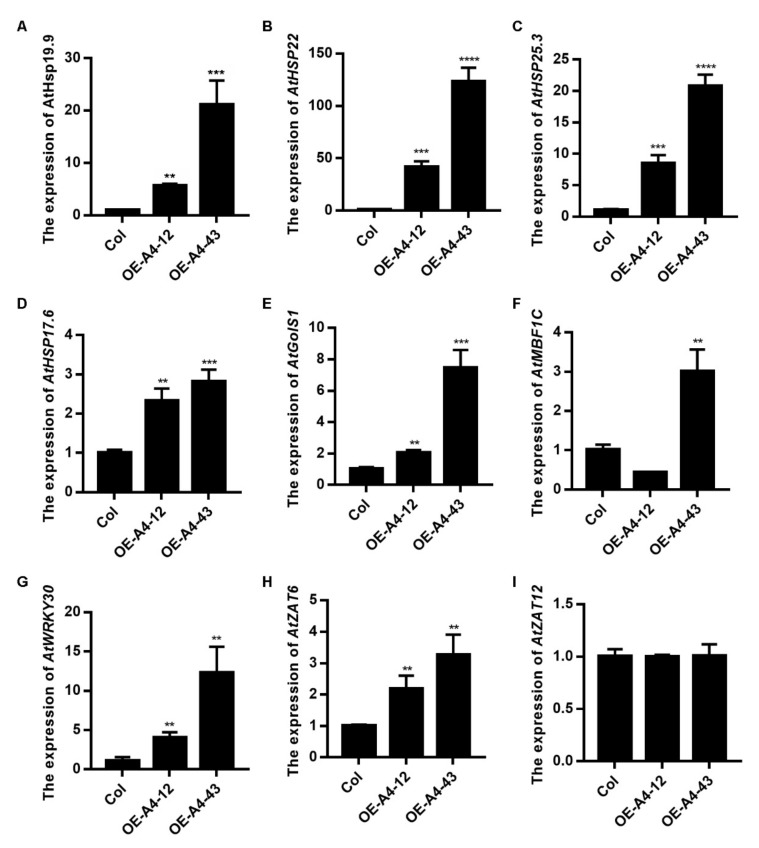
Expression of target genes was examined in *LlHsfA4* transgenic plants. (**A**–**I**) showed the expression levels of 9 target genes..The mRNA level of target genes was detected in wild type (Col) and transgenic lines using qRT-PCR. The expression value of these genes in Col was set as1 for comparison and *AtACTIN2* was used as an internal standard. 10-day-old seedlings were collected for RNA extraction. Three independent experiments were performed, each with three technical replicates, showing one experiment result. *t*-test analysis of variance was employed to identify treatment means that differed statistically. Samples with different letters are significantly different: ** *p* < 0.01, *** *p* < 0.001, **** *p* < 0.0001, Unmarked means non-significance.

**Figure 8 ijms-23-00572-f008:**
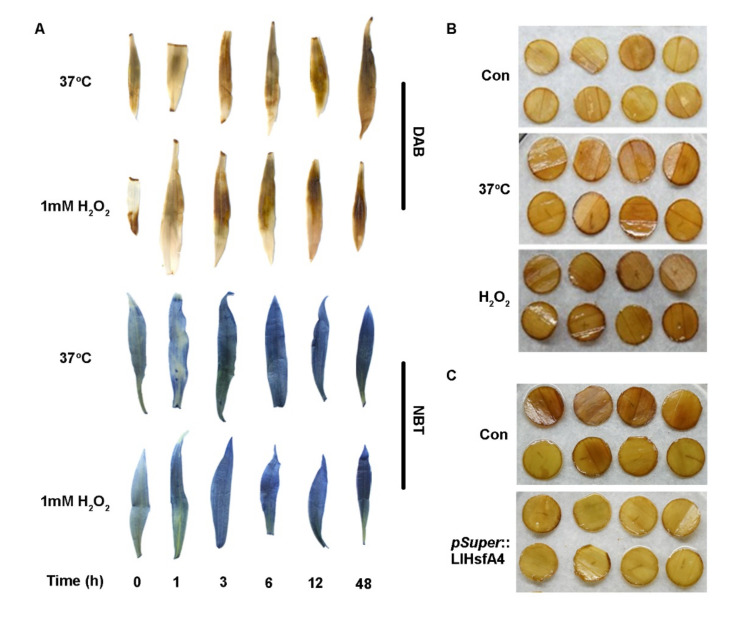
Content of ROS detected in lilies. (**A**) Leaves of ‘White Heaven’ tissue culture seedlings were stained by DAB staining or NBT staining under heat (37 °C) and H_2_O_2_ treatment. (**B**) Discs from mature leaves of cut flowers ‘Siberia’ were stained by DAB staining under heat (37 °C) and H_2_O_2_ treatment. (**C**) Discs from mature leaves of cut flowers ‘Siberia’ were stained by DAB staining under transient expression with *LlHs*fA4 under heat (37 °C) treatment.

**Figure 9 ijms-23-00572-f009:**
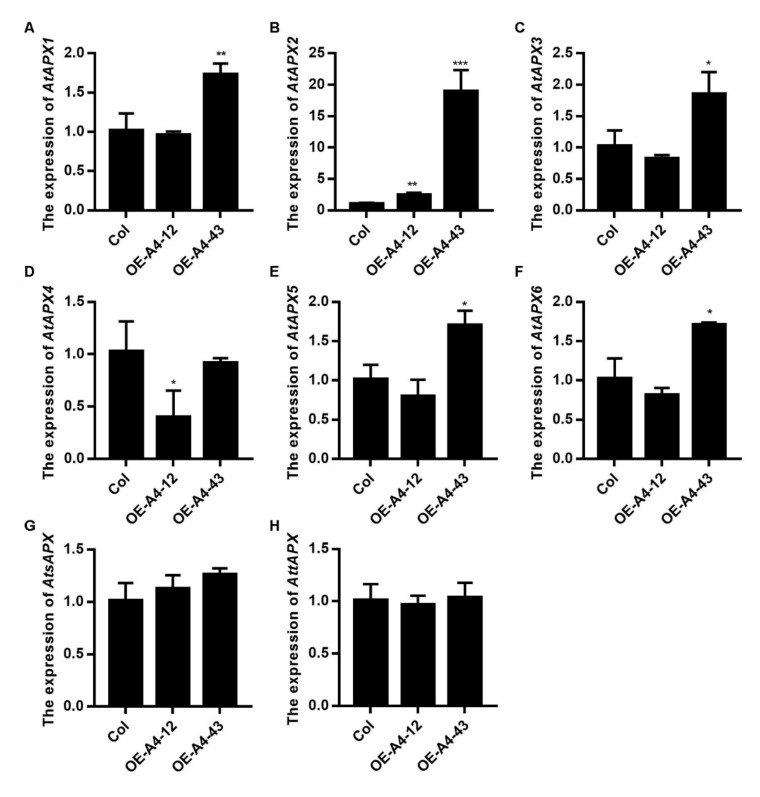
Expression of *AtAPXs* was examined in *LlHsfA4* transgenic plants. (**A**–**H**) show the Expression levels of 8 *AtAPXs*. The mRNA level of *AtAPXs* was detected in wild-type (Col) and transgenic lines using qRT-PCR. The expression values of these genes in Col were set as 1 for comparison and *AtACTIN2* was used as an internal standard. 10-day-old seedlings were collected for RNA extraction. Three independent experiments were performed, each with three technical replicates, showing one experiment result. *t*-test analysis of variance was employed to identify treatment means that differed statistically. Samples with different letters are significantly different: * *p* < 0.05, ** *p* < 0.01, *** *p* < 0.001, Unmarked means non-significance.

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
