# Peer review of "The Heat Stress Transcription Factor LlHsfA4 Enhanced Basic Thermotolerance through Regulating ROS Metabolism in Lilies (Lilium Longiflorum)"

_ijms, 2022, doi:10.3390/ijms23010572_

Round 1
Reviewer 1 Report
This manuscript by Wang et al., isolated HsfA4 from Lily (Lilium Longiflorum) using RACE method and performed homology comparison and phylogenetic analysis. LlHsfA4 is mainly expressed in leaves and is induced by heat stress and H2O2. It has transactivation activity and is located in nucleus and cytoplasm. The authors found that heat stress could increase H2O2 content in lily leaves and reduced H2O2 accumulation in transgenic plants.
Overall I have few concerns:
Figure 1 B What is the homology of LIHsfA4 with HSFs in other organisms like Nematostella which has 3 HSFs, yeast and humans?
Figure 5B Yeast one hybrid has been performed. Can the authors perform two hybrid with Hsp70/Hsp90 to test whether there is any effect/interaction?
Author Response
Please see the attachmen

Reviewer 2 Report
The work identified a HsfA4 homolog in lily plant, analysed phylogeny, domains and its expression in different tissues and abiotic stress conditions including heat and oxidative stress. The paper further shows that LlHsfA4 is nuclear and has transactivation domain based on a Y1H assay. It is suggested that LlHsfA4 has a defence role upon basic heat stress, but not during acquired heat stress; overexpressor lines accumulate higher level of HSP- and APX2 transcripts putting forward the hypothesis that LlHsfA4 may be involved in basic thermotolerance through ROS metabolism pathway.
Major concerns:
- As the basic thermotolerance has been done in a warm/hot environment, I suppose it was done in a growing cabinet; is such cases the HS regime is likely to test acquired thermotolerance, as the temperature sensing of plants is gradual. To do a BT test they need to apply the HS in a water bath on seedlings on plates (as widely done in the literature)
- Arabidopsis HsfA4 was shown to be involved in salt stress (Andrasi et al., 2019, JXB or others). They need to clarify the role of LlHsfA4 on salt stress (1-200 mM) to show whether indeed the gene/protein is indeed an ortholog of Arabidopsis counterpart.
- To prove that LlHsfA4 is involved in ROS metabolism, please test sensitivity of OX plants in different oxidative stress conditions as well.
- For nuclear localisation, pőlease introduce a staining control for visualisation of nuclei (e.g, DNA dye).
- In Figure 6 they need to do a time series of BT test (ideally in water bath); I would expect much clear differences.
- Upon checking the target genes in LlHsfA4-overexpression lines, a number of HSPs are induced (in accordance with the basal thermotolerance) but the evidences are not clear for oxidative pathway players (e.g. ZAT12 was described as being involved in oxidative pathway, therefore it is expected that it will change in the overexpressors, which is not the case). Additionally, there are huge differences (both qualitative and quantitative; transcript level changes sometimes are in opposite directions e.g. MBF1c or Cat1, Fig 7 and S3 respectively).
- Please check amount of superoxide using NBT staining to add to DAB staining (Figure 8).
- (minor) For a better quantification compare the amounts of GUS RNA (Fig. 3)
- (minor) Please make a better measurement of D1 construct (Fig 5), as it is expected to retain transactivation capacity (like D2); (otherwise, I would think of it as having a repressor region).
Overall, it seems that indeed by expressing the activator LlHsfA4 improves the thermotolerance of plants in a heterologous system. However, there is no prove that (i) this gene/protein is a bona fide ortholog of Arabidopsis counterpart A4 involved in salt stress tolerance or (ii) is an anti oxidative factor.
Round 2
Reviewer 1 Report
All my questions have been answered in the revised manuscript. I recommend the article for publication.
Author Response
Thanks for your comments.
Reviewer 2 Report
The authors have made several efforts, and improvement of the study can be observed. However, my previous concerns were not addressed deeply enough regarding presentation and scientific soundness. Below please find my comments. Overall, all my suggestions and comments were formulated to help the authors to aid a much higher quality for the publication. In the present form I cannot advice publication of the study.
- Molecular cloning and sequence alignment OK; Optional: would be nice to publish the genomic sequence in the supplementary.
- Tissue specific expression: pls indicate significance at bulbs (or ns!)
- Temperature course experiment: indicate significance at 42C (or ns!); ROS-induction fig captions has a 48h timepoint, pls add or delete! Please add a salt-induction assay! Would be important as you have a phenotype later!!
- Optional: pls add the promoter sequence in the supplementary, indicating the short TF binding motifs within, pls add the time and temperature of heat treatment to the fig 3B caption
- Please include the ref for the nuclear localization of NFYA4 in the main text
- In Fig 4B the bright pictures are very dark, therefore difficult to see the cellular structures. Pls brighten them!
- line 224 rephrase sentence! ”all the yeast that transformed plasmid could grow... ”
- line 237 rephrase sentence! „...and performed a yeast one hybrid system.”
- line 240: pls rephrase the conclusion „transcriptional activation region was located between D2 and D3” D2 and 3 are the names of the constructs!
- lane 262: why a super promoter was used instead of own promoter, specially as own promoter was found to be heat responsive?? pls explain
- heat treatment has to be done in water bath! That is a must for testing BT. Plates should not be wrapped in aluminium foil since this will impede heat transfer. Plastic agar plates should be simply flown on the surface of water in a water bath. For constant and reliable temperature an aquarium circulating pump can be employed (this is a small and very cheap device). One of the lines does not show significant differences, therefore the test needs to be repeated in several bio reps to be able to show clear differences, as this is a central conclusion of the work!
- Fig 7 indicate significance or non-significance („ns”)
- Fig 8 pls do NBT staining also for disc in absence and presence of HsfA4-ox construct.
- lane 327: expression of APX2 was higher in the two transgenic lines (fig 9), however significant difference was not found for line 12. Conclusion is not precise. Pls redo the measurements and indicate significance or non-significance!
- overall the indication that LlHsfA4 acts to modulate BT through ROS pathway is very weak, since ROS pathway transcripts are not significantly changed en masse! To show HsfA4 as an antioxidative agent please test resistance of OX lines to oxidative damage by measuring lipid peroxidation, APX activity or other. Combine the data from Fig S4 with Fig 8, as they are relevant for the readers!!
- As AtHsfA4 was shown to be involved in salt stress resistance, pls include the salt stress results in the main text. Pls quantify the difference between wt and HsfA4.ox lines by measuring some if not all the following parameters like fresh weight, chlorophyll content, root length, survival. This would be an important addition to the work.
- lane 363: subcellular, pls correct
- lane 370: have you tested LlHsfA4 promoter induction in response to zinc? Please include in supplementary, or correct discussion since is misleading.
- lane 373 and 377: conclusions are overstated! promoter may extend in the transcribed region. The fact that in your assay the promoter fragment in the transient assay is not salt-responsive does not exclude the possibility that the gene is not salt responsive; please rephrase, see also my comment 3., especially as you have a phenotype!
- lane 429: incorrect conclusion „... APXs did”, APXs were induced in only 1 of the lines, in the other were unchanged or rather repressed. Pls test APX activity in OX lines exposed to salt and BT to show biological relevance.
- lanes 440-442: I did not understand why APXs’ promoter was wanted, and how it could be relevant for the present study?
- Lane 449: the discussion is misleading and incorrect: „Moreover, HsfA4 could can bind to the promoter of ZAT12 to activate ZAT12 expression [44]. These reports were are consistent with our result that the expression of ZAT6 and APX2 were was higher in LlHsfA4 overexpression lines than in the wild type (Figure 7), indicating that LlHsA4 could regulate ROS degradation and is located upstream of the ROS signals.” Notably ZAT12 was found to be UNCHANGED, See fig 7I.
In summary, it seems that overexpression of the LlHsfA factor can enhance stress tolerance of ox lines in a heterologous system, however I was not convinced that the present factor is the functional ortholog of AtHsfA4 and it is still unclear what roles it may have under heat, salt or oxidative stress conditions. (As an additional note, an own promoter driven LlHsfA4 construct could have been used for complementation of hsfa4 arabidopsis (done in parallel to OX lines) and test downstream targets involved in heat, salt, ROS and other stresses. However, I do understand that this would require half year to perform, therefore, it is not a demand).
Dear Editor; please provide enough time for the authors to complete the research properly.
Author Response
Please see the attachmen

Round 3
Reviewer 2 Report
The manuscript ha been improved considerably. What may be the role of LlHsfA4 is still a matter of debate. Salt stress response seems to be an important feature, however not studied in details. The final evidence that LlHsfA4 regulates stress response through ROS pathway is lacking.